# Protection Strategies Against Palmitic Acid-Induced Lipotoxicity in Metabolic Syndrome and Related Diseases

**DOI:** 10.3390/ijms26020788

**Published:** 2025-01-18

**Authors:** Zeltzin Alejandra Ceja-Galicia, Carlos Leonardo Armando Cespedes-Acuña, Mohammed El-Hafidi

**Affiliations:** 1Departamento de Biomedicina Cardiovascular, Instituto Nacional de Cardiología Ignacio Chávez, Mexico City 14080, Mexico; alejandra.ceja@cardiologia.org.mx; 2Departamento de Ciencias Básicas, Facultad de Ciencias, Universidad del Bio Bio, Avenida Andres Bello 720, Chillan 3780708, Chile; ccespedes@ubiobio.cl

**Keywords:** antioxidants, lipid droplets, lipotoxicity, metabolic syndrome, obesity, oxidative stress, palmitic acid

## Abstract

Diets rich in carbohydrate and saturated fat contents, when combined with a sedentary lifestyle, contribute to the development of obesity and metabolic syndrome (MetS), which subsequently increase palmitic acid (PA) levels. At high concentrations, PA induces lipotoxicity through several mechanisms involving endoplasmic reticulum (ER) stress, mitochondrial dysfunction, inflammation and cell death. Nevertheless, there are endogenous strategies to mitigate PA-induced lipotoxicity through its unsaturation and elongation and its channeling and storage in lipid droplets (LDs), which plays a crucial role in sequestering oxidized lipids, thereby reducing oxidative damage to lipid membranes. While extended exposure to PA promotes mitochondrial reactive oxygen species (ROS) generation leading to cell damage, acute exposure of ß-cells to PA increases glucose-stimulated insulin secretion (GSIS), through the activation of free fatty acid receptors (FFARs). Subsequently, the activation of FFARs by exogenous agonists has been suggested as a potential therapeutic strategy to prevent PA-induced lipotoxicity in ß cells. Moreover, some saturated fatty acids, including oleic acid, can counteract the negative impact of PA on cellular health, suggesting a complex interaction between different dietary fats and cellular outcomes. Therefore, the challenge is to prevent the lipid peroxidation of dietary unsaturated fatty acids through the utilization of natural antioxidants. This complexity indicates the necessity for further research into the function of palmitic acid in diverse pathological conditions and to find the main therapeutic target against its lipotoxicity. The aim of this review is, therefore, to examine recent data regarding the mechanism underlying PA-induced lipotoxicity in order to identify strategies that can promote protection mechanisms against lipotoxicity, dysfunction and apoptosis in MetS and obesity.

## 1. Introduction

Dietary habits characterized by a high intake of carbohydrates and saturated fats, in combination with a sedentary lifestyle, related to the incidence of metabolic syndrome (MetS) and obesity, are risk factors for cardiovascular diseases and type II diabetes. The high-carbohydrate diet exhibits a stronger association with the accumulation of adipose tissue compared with a high-fat diet; furthermore, the restriction of dietary carbohydrates is related to a significant reduction in body weight compared to a low-fat diet in overweight and MetS patients [1]. Moreover, the reduction in carbohydrate uptake decreases blood pressure and triglyceride (TG) concentrations and increases high-density lipoprotein (HDL) levels in overweight individuals with MetS [2,3]. Thus, a low-carbohydrate diet reduces the prevalence of MetS and offers several potential benefits for the treatment of obesity and type II diabetes in many people, even if they do not lose weight [4,5,6]. The high-carbohydrate diet is well known to enhance de novo lipogenesis and increases the biosynthesis of palmitic acid (16:0, PA), the major saturated fatty acid, corresponding to 20 to 30% of total fatty acids in the human body. PA metabolism is tightly regulated with the endogenous synthesis via de novo lipogenesis (DNL) and dietary intake [7]. Palmitic acid serves as a major substrate for myocardial metabolism, influencing energy production. However, depending on the concentration and exposure duration, PA can potentially affect energy production and cardiovascular tissue health. Therefore, the balance between low and excessive exposure to palmitic acid plays a critical role in cell health and death due to lipotoxicity [8,9]. Moderate PA levels support cardiac health, whereas excessive concentrations can trigger detrimental metabolic pathways, highlighting the need for a balanced dietary intake to maintain cardiovascular lipid homeostasis. PA at high concentrations promotes lipotoxicity that results in cell dysfunction and death in many cell types, including skeletal muscle cells, liver cells, β-cells and cardiomyocytes [10]. PA induces lipotoxicity through several mechanisms involving mitochondria dysfunction, endoplasmic reticulum (ER) stress, oxidative stress, inflammation and cell death through apoptosis [11]. In pathophysiological conditions of obesity, insulin resistance and MetS to store PA in lipid droplets (LDs) as triglycerides in adipose and non-adipose tissues such as the liver, skeletal muscle and heart is considered as an endogenous protection strategy to protect cells from excessive PA. The utility of LDs in providing energy for metabolic processes and membrane biosynthesis has been extensively characterized in adipocytes, where triglyceride hydrolysis enzymes such as adipose triglyceride lipase (ATGL) and hormone-sensitive lipase (HSL) activities depend on feeding, starvation and hormonal status [12,13]. In obesity and MetS, a chronic increase in triglyceride biosynthesis and accumulation in LDs within both adipose and non-adipose tissues results in enhanced FFA release that promotes oxidative stress and endoplasmic reticulum stress, thereby increasing inflammation, insulin resistance, diabetes and cardiovascular diseases [14].

To attenuate the lipotoxic effect of PA, some strategies have been addressed. As saturated and monounsaturated fatty acids differ significantly in their contributions to lipotoxicity, it appears that monounsaturated, such as palmitoleic and oleic, acids have been found to protect against PA-induced lipotoxicity in cell culture and rodent models by enhancing whole body insulin sensitivity, stimulating insulin secretion by β-cells, increasing hepatic fatty acid oxidation and improving the blood lipid profile [15,16]. Furthermore, oleic acid has been found to protect against PA-induced lipotoxicity, by channeling the PA and oxidized PUFA into LDs as a TG within the cytosol of hepatocytes, skeletal muscle and pancreatic β-cells in an obesity rat model [17,18]. Nevertheless, the oxidative stress linked to PA-induced lipotoxicity and the exposure of food lipids to high temperatures and oxygen that harm human health suggest the need for antioxidant protection strategies against PA-induced lipotoxicity. Therefore, this review aims to discuss the recent findings concerning the mechanisms involved in PA-induced lipotoxicity, as well as to examine various endogenous and exogenous interventions aimed to protect cells from mitochondrial dysfunction, oxidative stress, ER stress and cell death and to protect lipid-rich foods from oxidation and subsequent health degradation.

## 2. Palmitic Acid Biosynthesis in Metabolic Syndrome and Obesity

PA accounts for approximately 20 to 30% of total fatty acids in human and animal bodies [7,19]. PA is synthetized endogenously from the end-products of glycolysis (pyruvate from amino acids and from other fatty acids). As illustrated in Figure 1, PA biosynthesis starts at the glycolysis end-product via the transformation of pyruvate to citrate, subsequently undergoing a conversion into acetyl-CoA. This acetyl-CoA is converted by acetyl-CoA carboxylase (ACC) to malonyl-CoA, which serves as a substrate for fatty acid synthase (FAS) that catalyzes the biosynthesis of PA. Therefore, a high-carbohydrate diet not only induces hyperinsulinemia but also increases de novo lipogenesis, contributing to increased levels of PA and triglycerides in adipose and non-adipose tissues [20]. Furthermore, PA as a free fatty acid is also enhanced by high lipolytic activity in adipose tissue due to insulin resistance.

Adipose triglycerides lipases, hormone-sensitive lipases, are among the key enzymes in the hydrolysis of triacylglycerol, diacylglycerol and monoacylglycerol, producing FFAs and glycerol. FFA release and mobilization from adipose tissue to peripheral tissue are regulated by insulin levels, fasting states, feeding conditions and hormonal status [21]. Nevertheless, when the release of PA via lipolysis activity exceeds its subsequent beta-oxidation, its accumulation leads to several metabolic alterations, as well as cell dysfunction and death through apoptosis and necrosis [22]. The disruption of the homeostatic balance related to the uncontrolled endogenous biosynthesis of PA involves different physiopathological conditions such as MetS, obesity and related metabolic diseases. In the liver, the stimulation of de novo lipogenesis through hyperinsulinemia involves the transcription factor sterol regulatory element-binding proteins-1c (SREBP-1c), which up-regulates the enzymes that catalyze lipogenesis [23]. Lipogenesis is also stimulated by glucose through the activation of the transcription factor of carbohydrate regulatory element-binding proteins (ChREBP) [24]. Both SREBP-1c and ChREBP up-regulate different genes involved in fatty acid biosynthesis, including desaturation and elongation enzymes, as shown in Figure 1 [25,26]. However, under the conditions of a high sucrose intake, the accumulation of PA is partially prevented by its enhanced activation to palmitoyl-CoA by acyl-CoA synthetase and its unsaturation to palmitoleyl-CoA (16:1n−7) by Δ-9 desaturase. Palmitoyl-CoA can also be elongated to stearoyl-CoA (C18:0) by fatty acid-like family member 6 (ELOVL6) elongase, and it is further unsaturated to oleyl-CoA (18:1n−9) by stearoyl-CoA desaturase and, finally, incorporation into triglycerides by Acyl-CoA:diacylglycerol O-acyltransferase (DGAT), which catalyzes the final step in TG synthesis, converting diacylglycerol into triglycerides, which, in turn, are stored in lipid droplets [11,27]. During the development of MetS and obesity due to sucrose feeding, the stearoyl-CoA desaturase activity positively correlates with palmitic and palmitoleic acid and plasma TG concentrations [28,29,30]. The desaturation of PA to palmitoleic acid is considered as endogenous protection against PA-induced lipotoxicity. Therefore, some studies have proposed that the substitution of palmitic acid by monounsaturated acid should be beneficial for diabetic and related cardiovascular diseases [31,32]. In addition, losing weight decreases the hepatic TG content by decreasing de novo lipogenesis in patients with non-alcoholic fatty liver disease (NAFLD) and increasing circulating glucose and insulin [33].

In addition to de novo lipogenesis, PA comes from the diet, where it is found in very large quantities compared to other fatty acids. In physiological conditions, the postprandial PA from food is quickly metabolized, incorporated in the cellular membranes and/or used for energy expenditure, making it an important nutrient that must be consumed [7]. The esterified PA in food triglycerides and phospholipids is absorbed in the small intestine, packing in chylomicrons and, subsequently, in lipoproteins from the liver, then being distributed to the organs through the blood stream [34]. In the capillary, the lipoprotein lipase (LPL) hydrolyzes the fatty acids from the glycerol to be absorbed from the cell and normally esterified in adipose tissue or oxidized in other organs. Moderated levels of PA in the cell promote beta-oxidation and ATP production, principally in the heart, skeletal muscle and kidney, maintaining mitochondrial function and cellular homeostasis [35]. On the other hand, a high-fat diet rich in saturated fatty acids such as PA is associated with insulin resistance in adipose tissue and enhanced lipolytic activity due to insulin resistance and an increase in non-esterified fatty acids, including PA.

## 3. Palmitic Acid-Induced Lipotoxicity

As previously elucidated, de novo PA is typically esterified in phospholipids or triglycerides to avoid its deleterious effect. However, when the PA availability exceeds its esterification capacity, the free form of PA can trigger mitochondrial dysfunction, endoplasmic reticulum (ER) stress, oxidative stress and inflammation, which are involved in the lipotoxicity induced by PA and contribute to cell death through apoptosis. However, mitochondria ROS generation and ER stress are the major participating mechanisms and can be attenuated by several antioxidant treatments using cell culture and experimental animals of obesity and MetS.

### 3.1. Palmitic Acid-Induced Endoplasmic Reticulum (ER) Stress

Palmitate-induced cardiomyocyte dysfunction and β-cell death in type II diabetes involve the participation of ER stress as the primary site of oxidative protein folding catalysis mediated by oxidoreductin-1α and ERO-1β isoenzymes through an over-production of hydrogen peroxide (H_2_O_2_) [36]. ER stress induced by PA is associated with the unfolded protein response (UPR) as a compensatory mechanism mediated through the up-regulation of activating transcription factor 4 (ATF4), inositol-requiring enzyme 1 (IRE1) and C/EBP homologous protein (CHOP), involved in the proapoptotic signaling pathway to mitigate dysfunctional cells [37]. In the myocardium of patients with type II diabetes, the levels of ER stress markers such as ATF4 and CHOP are found to be elevated due to metabolic disturbances of PA overload [38]. Among the mechanisms proposed to explain the induction of ER stress and loss of β-cell viability in response to PA treatment, the protein palmitoylation is a post-translational modification that regulates protein localization, stability and activity. Indeed, PA-induced ER stress in β-cells has been described to be mediated by excessive protein palmitoylation [39]. In MetS and obesity, the high availability of PA and its activated form palmitoyl-CoA serves as the S-palmitoylation protein agent. The S-palmitoylation of protein is a reversible reaction catalyzed by palmitoyl protein acyl transferase (PPT) and acyl protein thioesterase (APT), which depend on PA availability. Some proteins involved in the lipid metabolism such as CD36, a fatty acid transporter, are activated via palmitoylation to enhance fatty acid accumulation within cells (for a review, see [40]). The palmitate-induced protein palmitoylation has been described for the serine palmitoyl transferase 2 (SPTLC2), the rate-limiting enzyme of de novo ceramide synthesis. SPTLC2 catalyzes the condensation of serine and palmitoyl-CoA, forming 3-ketosphinganine, which is, thereafter, transformed into ceramide in hepatocytes exposed to palmitic acid [41]. Furthermore, ER stress is related to the increased total ceramide content and, specifically, due to the increased ceramide C14:0 and ceramide C16:0 [41]. In muscles from mice fed a high-fat diet, ceramide accumulation is linked to insulin resistance [42], impacts liver homeostasis and induces metabolic disorders [43]. Additionally, inhibiting desaturase SCD or SREBF1/2 leads to increased dihydroceramide levels, indicating their involvement in ceramide biosynthesis [44]. In primary mouse hepatocytes exposed to PA, cell death via apoptosis can be prevented using myriocin as an inhibitor of ceramide synthesis or by taurourso-deoxy-cholic to prevent ER stress in hepatocytes [44]. The inhibition of ceramide synthase effectively mitigates PA-induced β-cell ferroptosis [45]. Similarly, PA contributes to apoptosis through de novo ceramide synthesis in various cell types via oxidative stress mechanisms induced by the activation of the TLR4-ROS-p53 pathway [46] and the ERK1/2 signaling pathway, leading to increased levels of Bax, as well as a decrease in Bcl-2, which contributes to the apoptosis of H9c2 cells [47].

In addition, ceramides play a significant role in inducing apoptosis through other mechanisms, involving mitochondrial pathways and redox signaling. Ceramides bind to voltage-dependent anion channels (VDACs) 1 and 2 in mitochondria, which is crucial for triggering apoptotic pathways. The loss of VDAC2 significantly reduces ceramide-induced apoptosis [48]. In relation to redox signaling, ceramides enhance the expression of thioredoxin-interacting protein (Txnip), concomitantly reducing thioredoxin 1 (Trx1) activity, which subsequently leads to increased apoptosis [49]. The altered redox signaling induced by ceramide is also linked to a shift in the balance of superoxide dismutases (SOD) 1 and 2 [50]. Furthermore, the activation of ER stress by PA induces the release of calcium from the ER and results in elevated cytosolic calcium concentrations, thereby perturbing calcium homeostasis and facilitating the accumulation of reactive oxygen species (ROS), which, in turn, promotes the occurrence of ferroptotic cell death [51]. Thus, the disruption of calcium homeostasis through excessive Ca^2+^ release in the ER and the resulting mitochondrial Ca^2+^ overload increase oxidative stress and lipid droplet accumulation, which are essential in the progression of insulin resistance and other diabetic complications [52].

### 3.2. Palmitic Acid-Induced Mitochondrial Dysfunction

Palmitic acid-induced mitochondria dysfunction is mediated by several mechanisms, involving altered fatty acid beta-oxidation, ROS generation and the collapse of the mitochondrial membrane potential (MMP). All these mechanisms involve cytochrome c release, with caspase-3 activation being a key enzyme in the cell death process via apoptosis [53].

In cardiomyocytes, PA at high concentrations has been described to impair mitochondria energy metabolism by inhibiting fatty acid beta-oxidation, resulting in lipid accumulation and decreased ATP biosynthesis [54]. In addition, PA-increased ROS generation and -decreased ATP biosynthesis induce alterations in the expression of peroxisome proliferator-activated receptor (PPAR) α, δ and γ, PGC1α and UCP2 involved in fatty acid metabolism [54]. Thus, PPARs play a significant regulatory role in the mitochondrial biogenesis signaling pathway and fatty acid metabolism in diabetic cardiomyopathy. In response to PA treatment, the expression levels of PPARα and PPARγ were found to be decreased, while PGC-1α and UCP2 levels were found to be increased, suggesting complex interaction between lipid accumulation, ROS production and PPAR signaling [54].

Palmitic acid also down-regulates the expression of fusion genes (MFN1) and up-regulates the fission genes (FIS1), disrupting the balance between fusion and fission, leading to apoptosis and impaired oxidative metabolism [55]. Mitofusin2 (Mfn2), the inducer of the mitochondria fusion, directly interacts with carnitine palmitoyl transferase (Cpt1α), a fatty acid transporter, into mitochondria, via its GTPase domains, which seems to be essential for maintaining the activity of Cpt1α and promoting mitochondrial fatty acid beta-oxidation [56].

In macrophages, palmitic acid impairs the mitochondrial function by decreasing respiratory complex IV and succinate dehydrogenase activities and up-regulating the adipocyte fatty acid-binding protein (FABP4), which compromise fatty acid oxidation and increase ROS generation [57]. In addition, palmitic acid negatively impacts aconitase and isocitrate dehydrogenase (IDH) activities [58]. In summary, PA and its biologically active derivative, palmitoyl-CA, engage multiple biochemical pathways that result in elevated ROS production via the inhibition of mitochondrial complexes I and III and the stimulation of peroxisomal activity and protein kinase C (PKC) signaling pathways. Collectively, these mechanisms lead to increased ROS production in hepatic mitochondria and macrophages, which is associated with increased expression of proinflammatory cytokines, thereby intensifying oxidative stress [59].

In an H9C2 cell model of lipotoxicity induced by high levels of PA exposure, the externalization of mtDNA in the cytosol is a result of the PA-induced over-production of mitochondrial ROS, promoting diabetic cardiomyopathy and cell death via apoptosis [60,61]. Indeed, PA also enhances caspase-3, leading to programed cell death and DNA fragmentation, and alters cell signaling in HepG2 cells under glucolipotoxic conditions [53,62]. PA-induced apoptosis is also accompanied by autophagy that, when blocked, exacerbates cell damage [63].

In a model of metabolic syndrome and obesity induced by high-sucrose consumption, a change in membrane phospholipids, especially cardiolipin (CL), exclusive of the mitochondrial inner membrane (MIM), was observed [64]. Its four-chain polyunsaturated fatty acid and conical structure enable it to interact with several proteins, enhancing the communication between them and regulating mitochondrial dynamics and homeostasis. For example, CL concentrates near the membrane contact sites of proteins such as OPA1, which are essential for membrane fusion and remodeling [65]. The high availability of PA [64] promotes its incorporation into CL molecules, and the enrichment of CL with PA modifies its tridimensional structure and, therefore, mitochondrial function. This modification contributes to the pathogenesis of metabolic and cardiovascular disorders via various molecular mechanisms, mainly those associated with lipid metabolism, mitochondrial architecture and respiratory efficiency [66]. These may also be related to decreased mitochondrial membrane fluidity and the stability of respiratory complexes [67].

Under conditions of lipid saturation stress, CL is crucial for maintaining the ultrastructure of the MIM, as shown in yeast models [68]. It has been reported that PA, when incorporated into CL molecules, alters the composition and potentially affects mitochondrial function and cell proliferation through the activation of the NOD-like receptor (NLR) signaling pathway [69,70]. Prolonged exposure of mitochondria to PA increases the production of ROS by inducing oxidative stress and damaging cardiomyocytes [8,71].

On the other hand, high levels of PA-induced oxidative stress within cardiomyocytes modify lipid metabolism, adversely influencing CL composition [8,9] and impacting mitochondrial biogenesis and dynamics [9]. Indeed, unsaturated fatty acids in cardiolipin are particularly susceptible to oxidative damage, leading to altered fatty acyl profiles [72]. Altered cardiolipin composition due to oxidative stress is linked to increased apoptosis in cardiomyocytes [71]. However, oxidative stress triggers remodeling processes that replace cardiolipin oxidized fatty acids with non-oxidized polyunsaturated fatty acids, further compromising mitochondrial function [72].

The loss of CL remodeling enzymes results in altered structural integrity of the MIM, such as the accumulation of monolyso-cardiolipin, interacting with membrane-associated proteins and altering the activity of these proteins, thus affecting mitochondrial dynamics [65]. CL remodeling is essential for adapting to different lipid environments and is crucial for mitochondrial health [68]. Unlike palmitic acid, which exerts adverse effects on cardiolipin composition and overall cellular vitality, specific fatty acids, including oleic acid, have the ability to prevent these negative consequences. This finding suggests a complex interplay between different dietary lipids and their cellular implications [69]. Thus, CL remodeling could be induced by the presence of a high PA concentration or because of an oxidative process to protect MIM against ROS. Such complexity points to the need for further research into the role of cardiolipin in various physiological and pathological contexts.

### 3.3. Palmitic Acid-Induced Inflammation

PA is well known to induce inflammatory responses, with disturbances in glucose metabolism and insulin resistance [73]. PA induces ROS inflammation via two main pathways: the production of superoxide anion during its metabolism and its direct accumulation in cells as diacylglycerols (DAG). This leads to the activation of protein kinase C, which phosphorylates and activates IKK, subsequently enhancing NF-kB activation [74]. This is the most approved theory; PA induces an indirect activation of NF-κB through the increased ER stress and mitochondria ROS production. Nevertheless, PA could directly participate in the inflammatory process. It has been hypothesized that, in macrophages, PA could induce this process though direct interaction with Troll-like receptors (TRL), specifically 4, and the differentiation cluster 36 (CD36) complex or with the interaction of the free fatty acid receptor (FFAR1/GPR40), enhancing the NF-κB inflammatory response through the PI3K/PKB/AKT or PLC/PKC/MAPK signaling pathways [73].

Also, it has been described that, in addition to TRL4, TLR6 and 2 could participate in the PA activation of the inflammasome NLPR3 and interleukin 1ß via NF-κB and through the activation of the PI3K/AKT/mTOR signaling pathway [75,76]. On the other hand, the PA activation of TRL4 also induces resistin/insulin resistance in human neuroblastoma through the formation of the TRL4/MYD38/TRIAP complex [77]. The interaction between PA, the TRL family and the direct relationship of PA-induced lipotoxicity with inflammation and insulin resistance have been demonstrated. In addition, it has been described that PA decreases the anti-inhibitor of nuclear factor κB (IκBα), interrupting the interaction and cytoplasmic stabilization of NF-κB, and, thus, allowing for its translocation to the nucleus in primary human myotubes [78]. Thus, a high concentration of PA in obesity and metabolic syndrome could activate, directly and indirectly, the inflammatory process through different signaling pathways, exacerbating lipotoxicity and breaking cell homeostasis to develop insulin resistance.

On the contrary, it has been mentioned that PA could play a role in the anti-inflammatory process and glucose metabolism disorders through free fatty acid receptors FFAR1 or FFAR4/F-κB/KLF7 signaling pathway in epididymal white adipose tissue and livers from mice fed a high-fat diet [19]. This approximation is related to PA-induced long noncoding RNAs (PARAIL)/ELAV like RNA-binding protein 1 (ELAV1) interaction in macrophages, where the PARAIL down-regulation of cytosolic ELVA1 decreases the activity on NF-κB and the synthesis of proinflammatory cytokines [79]. However, PARAIL could be a protective cell mechanism in response to high concentrations of PA. These results could support the idea that PA-induced inflammation is not only related to its concentration but that the long duration of exposure to a high concentration plays a crucial role in this respect.

## 4. PA-Induced Lipotoxicity and Free Fatty Acid Receptor Intervention

Free fatty acid receptors (FFARs) are G protein-coupled receptors (GPRs), such as GPR40, GPR41, GPR43, GPR119 and GPR120, are expressed in almost all tissues and are activated by a wide variety of ligands, such as hormones, neurotransmitters, peptides, proteins, steroids and FFAs, mediating their effects through the intracellular signaling cascade [80]. In particular, FFAR1 (GPR40) and FFAR4 (GPR120) are activated by long-chain saturated and unsaturated fatty acids, whereas FFAR3 (GPR41) and FFAR2 (GPR43) are activated by short-chain fatty acids (SCFAs), mainly acetate, butyrate, and propionate [81]. Among the five members of the FFAR family, FFAR1, also known as GPR40, has been the most extensively studied in the β-cell. FFAR1, predominantly expressed in pancreatic β-cells, structurally possesses seven transmembrane domains and is activated by various medium- and long-chain FFAs (C12–C22), triggering a signaling cascade, which leads to increased intracellular calcium levels and potentiates glucose-stimulated insulin secretion (GSIS), managing blood glucose in diabetic patients [82]. The mechanism by which FFAs acutely potentiate GSIS is also through FFAR1 and Gα_q_ signaling, activating phospholipase C (PLC) that, in turn, hydrolyzes phosphatidylinositol-4,5-bisphosphate (PIP_2_) into inositol trisphosphate (IP_3_) [83]. The inositol trisphosphate targets its receptors at the ER membrane, promoting calcium release from ER stores, which results in the further elevation of cytosolic calcium, primarily initiated by glucose stimulation [83]. The DAG resulting from PIP_2_ hydrolysis directly participates in the potentiation of GSIS by targeting PKC and modulating insulin granule exocytosis [84,85]. Indeed, PLC inhibition and FFAR1 RNA interference block intracellular calcium mobilization and insulin secretion in islets from rosiglitazone-treated OLETF rats with an over-expression of FFAR1 [86]. In physiological conditions, PA-treated cells enhance mitochondrial respiration via the combined action of the intracellular metabolism of the fatty acid and Gαq-coupled FFAR1 signaling, suggesting a synergic action of the two pathways on insulin secretion [87].

On the other hand, the prolonged exposure of cells to PA disrupts this intracellular signaling, impairing beta-cell function and GSIS [88]. In type II diabetes, increased levels of PA induce the loss of the β-cell function due to the involvement of FFAR1 in crosstalk with mTOR-Akt and IRS-1, altering insulin signaling [89,90,91]. In hepatocytes, mTOR signaling is implicated in the cellular response to lipotoxicity by enhancing triglyceride secretion and reducing cell survival [92]. The selective inhibition of mTOR promotes fatty acid catabolism and suppresses lipogenesis, thereby protecting against non-alcoholic fatty liver in mice [93] and managing metabolic disorder related to obesity [94]. In addition, an excess of PA as an FFA may increase ROS generation related to lipotoxicity and alters the pathways of GSIS through FFAR1 [95]. Indeed, palmitic acid has been found to promote the generation of superoxide anion via the activation of NADPH oxidase and mitochondria, which correlates with a reduction in FFAR1-mediated signal transduction pathways, inhibiting insulin secretion, even when glucose concentrations are elevated [96]. Thus, these works were addressed to elucidate the signaling pathways by which FFARs mediate the effects of fatty acids on insulin secretion and lipotoxicity, which may lead to new therapeutic strategies for managing insulin resistance and related metabolic disorders.

Several strategies have been considered to improve FFAR signal transduction via the up-regulation of their expression and the stimulation of their activities. FFARs can be activated by natural agonists, such as oleic acid, linoleic acid and do-cosahexaenoic acid, which can be considered safe. In the case of synthetic agonists, which are considered much more effective than natural ones, only TAK-875, and not GW9508 or NCG75, which have been used in research involving cells in culture and experimental animals, has shown efficacy with no side effects [97]. TAK-875, a selective FFAR1 agonist, has been found to increase insulin secretion and improve glucose tolerance without causing hypoglycemia and other serious side effects in patients with type II diabetes for the first time, making it a potential new treatment for type II diabetes, as reported by Leifke et al. and Araki et al. [97,98].

In a recent study by Dragano et al. [99], it was described that FFAR1 activation increases energy expenditure and promotes the browning of subcutaneous white adipose tissue, thus contributing to weight loss. However, significant differences in the effects of FFAR1 activation can be found between obese and non-obese individuals, particularly in terms of energy balance and metabolic responses. In obese individuals, FFAR1 activation leads to increased energy expenditure and enhanced thermogenesis in brown adipose tissue, with a reduction in inflammation and endoplasmic reticulum stress in the hypothalamus [99]. In contrast, non-obese individuals may not benefit from these pronounced metabolic advantages, as their basic energy regulation mechanisms are different. In rat proximal tubular cells (NRK52E), the stimulation of FFAR1 by specific agonists mitigates the superoxide production and cytotoxic effects induced by PA in a dependent manner on Nrf2/HO-1 [100,101]. In β-cells, the up-regulation of the expression of FFAR1 at the mRNA through pioglitazone treatment, an insulin sensitizer, enhances the antioxidative stress protein levels and reduces PA-induced lipoapoptosis [102]. Unlike saturated FFAs, long-chain unsaturated FFA-induced FFAR1 over-expression protects rat pancreatic β-cells (INS-1) against lipotoxicity [103]. The FFAR1 overexpression induced by unsaturated FFAs is more related to the acute protection of FFA-activated FFAR1 than that of FFAs, mediating chronic β-cell lipotoxicity [104]. Furthermore, FFAR4, which is also stimulated by long-chain fatty acids, plays a crucial role in mediating a protective response against lipotoxicity; this effect is completely diminished in cells where FFAR4 is inactivated and in islets isolated from FFAR4 knockout mice that exhibit glucose intolerance and insulin resistance [105]. This observation implies that FFAR4 activation could potentially serve as a preventive and therapeutic target for the management of obesity and diabetes [105].

Hence, future research should address and elucidate the signaling pathways by which FFARs mediate the effects of fatty acids on insulin secretion and lipotoxicity, which could lead to new therapeutic strategies for managing insulin resistance and related metabolic disorders.

## 5. Palmitic Acid Reduced Insulin Sensitivity and Increased Vascular Dysfunction

In obesity and MetS, the excessive accumulation of PA enhances metabolic complications, leading to insulin resistance (IR) and type II diabetes [106,107]. Among the mechanisms of PA, reducing insulin sensitivity in several cell types, the influence of PA on the plasma membrane lipid organization may influence the amount and affinity of insulin receptors to insulin and the modulation of its phosphorylation [108,109,110]. In skeletal muscle cells, PA reduces glucose uptake by enhancing the serine phosphorylation of insulin receptor substrate-1 (IRS-1) and decreasing insulin-mediated Akt activation and glucose transporter 4 (GLUT4) translocation [111]. In pancreatic cells, PA affects GSIS through the phosphorylation of both IRS-1 in the serine residue S636/639 and Akt in S473 [89]. Moreover, PA decreases the expression of the insulin receptor 1ß (IR1ß) [89]. The high content of PA in the membrane of erythrocyte and hepatocytes makes it more rigid, with a decrease in the affinity of insulin to its receptor that results in decreased insulin sensitivity and hyperinsulinemia [108,112]. The insulin and glucose impairment induces type II diabetes development. According to this, reviews have shown that high-carbohydrate and high-fat diets contribute to lipotoxicity in pancreatic ß-cells, ultimately leading to disfunction attributable to oxidative stress and ER stress [113]. Also, the deficiency of depalmitoylation enzymes or increase in the palmitoylation process of diverse proteins activates insulin hypersecretion that causes stress to ß-cells, progressing to insulin deficiency [114]. The lipotoxicity process in diabetes can lead to diabetic cardiomyopathy through PA-induced ROS production and mitochondrial DNA release that activates the expression of proinflammatory factors [57].

PA is implicated in the development of cardiovascular diseases through several molecular mechanisms. These mechanisms primarily involve inflammatory responses, endothelial dysfunction and atherosclerotic plaque instability. In normal cardiomyocytes, palmitate alters calcium handling, reducing cytosolic Ca^2+^ transience and cell contraction, whereas in insulin-resistant cells, palmitate potentiates these parameters [115]. This dual effect is related to the increased mitochondrial production of ROS [115] and to the activation of the NADPH oxidase (NOX)/ROS signaling pathway, which decreases the phosphorylation of endothelial nitric oxide synthase (eNOS) at critical sites, impairing nitric oxide production in endothelial cells. These alterations are linked to increased oxidative stress and reduced vascular function. In addition, PA activates long-chain acyl-CoA synthetase-1, further promoting inflammation in endothelial cells via TLR pathways [116]. It binds to the TLR4 accessory protein MD2, leading to myocardial inflammatory injuries [117]. In type II diabetes, PA enhances atherosclerotic plaque vulnerability through macrophage delta-like ligand 4 (Dll4) signaling, which induces vascular smooth muscle cell senescence and reduces collagen synthesis and plaque stability [118]. In addition, high levels of PA impair the bioavailability of endothelial progenitor cells, crucial for vascular repair, via a PPARγ-mediated mechanism, which may exacerbate cardiovascular disease [119]. Therefore, PA at high concentrations influences the structural integrity of phospholipid membranes, thereby modulating insulin sensitivity and inducing cellular starvation. This produces changes in the cell homeostasis, making them vulnerable to oxidative stress and the inflammatory process, resulting in tissue injury and the development of diabetes or cardiovascular disease.

Although the lipotoxic effects of a high concentration of PA are well documented, short-term exposure to low-dose palmitic acid has been shown to improve mitochondrial function and oxidative status in cardiomyoblasts, suggesting that controlled exposure may enhance cellular resistance to mitochondrial dysfunction, oxidative stress, inflammation and insulin resistance [8]. Thus, the possibility that low doses of PA confer protective benefits suggests the need to explore dose–response relationships further in clinical settings. This duality could inspire dietetic recommendations and personalized therapeutic strategies.

## 6. Palmitic Acid-Induced Lipotoxicity Is Attenuated by Natural Antioxidants

The impact of mitochondria-targeted antioxidant MitoQ on the cardiac damage induced by obesity suggests crosstalk between PA-induced lipotoxicity and mitochondrial ROS generation [120]. In an in vitro model of liver steatosis induced by palmitic/oleic, chlorogenic and lipoic acid, the prosthetic groups of the pyruvate dehydrogenase complex and a potent antioxidant bioactive, respectively, reduced PA-induced lipotoxicity, improved mitochondrial membrane potential, and decreased ROS production and Bax expression, thereby reducing mitochondria-mediated caspase-dependent apoptosis in hepatocytes [121,122]. Between polyphenolic compounds, chlorogenic acid also attenuates PA-induced lipotoxicity through the activation of SIRT1 (silent information regulator1) in hepatocytes and suppresses the activation of the c-Jun NH2-teminal kinase (JNK) signaling pathway, inhibiting ER stress [123]. PA-induced ROS production, ER stress and death in liver cells were also prevented by luteolin and melatonin treatment, a known antioxidant, and anti-inflammatory compounds, through the up-regulation of the expression of antioxidant enzymes, i.e., heme oxygenase-1 and glutathione peroxidase [100,124].

Between coumarins, scopoletin, esculetin and umbelliferone are molecules with antioxidant and anti-inflammatory properties, which significantly reduce oxidative stress in PA-treated hepatocytes by decreasing mitochondrial superoxide production and increasing the expression of antioxidant genes, including Nrf2 and Gpx1, thus reducing ER stress by inhibiting the phosphorylation of JNK, a cell death signaling intermediate [125]. In the same way, highland barley tea extract, rich in polyphenols and flavonoids, represses PA-induced apoptosis and mitochondrial dysfunction in C2C12 myocytes by promoting the expression of Sirt3 through the phosphorylation of adenosine 5‘-monophosphate (AMP)-activated protein kinase (AMPK), which also activated SIRT3/FoxO3a to scavenge ROS in excess [126]. In summary, various antioxidants have been developed and tested to mitigate PA-induced lipotoxicity. However, further investigations into the efficacy and specificity of natural antioxidants in reducing oxidative stress and improving cellular health in models of metabolic MetS and obesity are needed.

PA-induced IR also involves intracellular mechanisms sensitive to oxidative stress and ROS generation. Several natural extracts from higher plants, such as phenolic and flavonoid compounds with antioxidant properties, have been tested and shown to be protective against PA-induced IR in cell cultures [127,128]. For example, Epigallocatechin gallate and resveratrol stimulate glucose uptake mediated by the PI3K-independent mechanism that involves SIRT1 in skeletal muscle cells [129] and improve PA-induced IR in C2C12 cells through the DNA damage-inducible transcript 4 (DDIT4)/mTOR/IRS-1/PI3K/AKT/GLUT4 signaling pathway [130]. Epigallocatechin gallate also improves PA-induced IR and reduces inflammatory and oxidative stress levels through NF-κB, tumor necrosis factor-α, interleukin-6, p53, ROS and by enhancing SOD and glutathione peroxidase [129]. In HepG2 cells, theaflavins, the bioactive polyphenols in black tea, protect against PA-induced insulin resistance by up-regulating total and membrane-bound GLUT4, increasing the phosphorylation of Akt at Ser473 residue, and decreasing the phosphorylation of IRS-1 at Ser307 [131]. Other natural polyphenols, such as salvianolic acid A, extracted from *Salvia miltiorrhiza Bunge* protect against PA-induced lipotoxicity in livers from high-fat–high-carbohydrate (HFCD)-fed mice and in hepatocytes through a TLR4/MAPKs-mediated mechanism [123,124]. Other natural phenolic compounds, such as curcumin, quercetin, resveratrol and sulforaphane, widely known as natural Keap1-Nrf2 activators, react with critical thiol groups of Keap1, leading to the release and activation of Nrf2, enhancing cellular protection against toxicity, which could mitigate PA-induced oxidative stress through the expression of antioxidant enzymes [132]. In summary, various antioxidants have been developed and tested to mitigate PA-induced lipotoxicity. Notably, the main mechanism is related to the activation of sirtuins, Akt and AMPK, with polyphenols being the most effective. Also, it appears that PA-induced lipotoxicity is mainly related to oxidative stress and that antioxidants diminish this process. However, lipotoxicity is not totally attenuated, suggesting that PA induces lipotoxicity through other signaling pathways. Further investigations on the efficacy and specificity of natural antioxidants in reducing oxidative stress and improving cellular health in models of metabolic MetS and obesity are needed to provide an appropriate therapeutic target that could help improve antioxidant therapy.

## 7. PA-Induced Lipotoxicity Is Attenuated by Lipid Droplet (LD) Functionality

The biogenesis of LDs from ER is induced in cells exposed to excessive amounts of lipids, nutrients and oxidative stress [133]. However, several other conditions, such as energy and redox imbalances, limit LDs and accelerate lipophagy, a part of autophagy that maintains cellular homeostasis, by removing damaged proteins and lipids [134,135]. In normal beta-cells, triacylglycerol accumulation in LDs is considered a protective mechanism to prevent the cellular rise of toxic free fatty acyl moieties that serve as inducers of cell death [136]. Thus, the biosynthesis and turnover of LDs within cells are precisely regulated to maintain the balanced lipid distribution and allow for cellular adaptation during saturated fatty acid stress [137].

In non-adipose cells, cytosolic LDs store neutral lipid triacylglycerols, and, when energy is required, they are hydrolyzed to FFA and glycerol as an oxidative substrate for ATP biosynthesis. Lipid droplets, however, have been described to form dynamic contact sites with mitochondria and peroxisomes, facilitating the transfer of fatty acids for beta-oxidation and energy production [138]. These interactions will depend on the metabolic state of the cell and will vary according to cell and tissue type. While lipid droplets are essential for efficient energy utilization, their dysfunction can also lead to metabolic disorders, such as obesity and diabetes, underscoring the importance of new research that will enable us to understand their biogenesis and interactions with other organelles in order to develop therapeutic strategies for lipid-associated diseases.

In primary cultures of cardiomyocytes from adult rats, the protective mechanism against PA-induced apoptosis involves channeling PA into triglyceride pools, which accumulate in LDs, decreasing DNA fragmentation and cell death through apoptosis [139,140]. The storage of harmful PA into inert triglyceride (TG) in LDs within skeletal muscle myotubes has been evidenced by the rapid incorporation of the fluorescent PA analogue into arachidonic acid-driven TG droplets [17]. Adipocyte LDs have a great capacity to store the exogenous PUFA delivered from the diet as TGs to avoid their accumulation within the phospholipid membrane that render cells more susceptible to oxidative stress [141]. In addition, LDs protect against lipid peroxidation, control the availability of energy for muscle function, modulate mitochondrial activity and enable the transfer of fatty acids between organelles, but they influence lipotoxicity in metabolic diseases [142]. In cardiomyocytes, lipids that exceed the cell capacity for LD storage induce non-ischemic and non-hypertensive cardiomyopathy, known as lipotoxic cardiomyopathy [143].

Lipid droplets have also been reported to control PUFA storage in triglycerides in order to reduce membrane lipid peroxidation and, in turn, may act as antioxidant organelles to preserve the cell function and prevent cell death due to ferroptosis [135,144]. Consequently, LDs play a crucial role in regulating PUFA transport and facilitating the management of membrane lipid peroxidation, ferroptosis and lipid mediator signaling [145]. The presence of LDs within renal cells suggests significant potential for the diagnosis and enhanced comprehension of chronic kidney disease, due to the concurrent presence of oxidized lipid species, such as triglycerides, phosphatidylcholines and phosphatidylethanolamine hydroperoxides, in addition to the dysregulation of lipid metabolic pathways [146]. The protective effect of LDs is also associated with the replacement of PUFAs with MUFAs that are more resistant to lipid peroxidation. Indeed, the treatment of cells with exogenous MUFAs reduces the sensitivity of plasma membrane lipids to oxidation and activates acyl-coenzyme A synthetase long-chain family member 3 (ACSL3) involved in LD formation and the insertion of MUFA in membrane phospholipids [147]. Additionally, PUFAs, including linoleic acid obtained from nutritional intake, are transferred from cellular membranes to the nucleus of LDs, where they are more protected from peroxidation, thereby highlighting the antioxidant functions of LDs [145]. Thus, the endogenous formation of LDs diminishes PA lipotoxicity, reducing the presence of FFAs, avoiding interaction with lipolysis enzymes and, therefore, the excessive production of oxidative phosphorylation substrates that can produce ROS. Also, it could prevent increased phospholipids enriched with saturated fatty acids in the membrane and the palmytoylation of several proteins, preserving cellular homeostasis.

## 8. Natural Antioxidants and PA-Induced Lipid Droplets

The intracellular accumulation of TG and its oxidized species (TG hydroperoxides) in LDs triggered by PA in hepatocytes can also be regulated by polyphenols and flavonoids with antioxidant activities, decreasing the release of lactate dehydrogenase, an important indicator of cytotoxicity [148,149]. In HepG2 cells with oleic acid and PA-induced lipid accumulation, baicalein, a flavonoid derived from *Scutellaria-baicalensis,* reduces the production of TG, total cholesterol and lipid droplets by decreasing the expression of SREBP1, as well as the lipogenic enzymes fatty acid synthase (FAS) and stearoyl-CoA desaturase-1 (SCD1) [150]. The addition of N-acetylcysteine (NAC) as an antioxidant to the retinal pigment epithelium reduces oxidative stress mediated by lipid metabolism, which plays an important role in lipid droplet (LD) accumulation and cellular senescence induced by high glucose level [151]. On the other hand, exercise training elevates the markers of lipid droplet dynamics such as the perilipin proteins and triglyceride lipases ATGL and HSL, as well as mitochondrial efficiency, improving lipid turnover and reducing lipotoxicity [152].

Thus, LDs play an important role in the regulation of cellular stress, suggesting their application as safe and effective biomaterials to reduce cellular stress and lipotoxicity. Therefore, the generation of stable and pure artificial LDs coated with perilipins, lipase regulatory proteins, has reduced hydrogen peroxide-induced oxidized lipid species and alleviated cellular lipotoxicity caused by excess FFAs [142,153].

In summary, LDs are usually protective in many diseases; however, the long-term accumulation of and uncontrolled increase in LDs in chronic diseases such as obesity and MetS can exacerbate disease progression [154]. It is, therefore, necessary to understand the lipid–protein interactions of physiological and pathological LDs in order to better understand their transition from beneficial to detrimental in different pathological contexts. In this case, investigating the sites of contact of lipid droplets with other organelles in healthy and diseased states will contribute to elucidating the mechanisms of disease progression and, thus, the development of therapeutic strategies based on the biogenesis and function of lipid droplets. Therefore, more studies are needed to clarify how lipid droplet dynamics can be manipulated to enhance their protective effects against lipotoxicity and their potential as therapeutic targets in cellular stress regulation.

## 9. Antioxidant Strategies to Prevent Lipid Peroxidation

Mitochondrial PA beta-oxidation has been described as a source of superoxide anion (O_2_**^−^**)(O_2_**^−^**), which, in turn, is protonated to generate the perhydroxy radical (HOO.) that promotes the lipid peroxidation of PUFA [155,156]. In order to attenuate lipid oxidation in the human body and food, several effective strategies using natural antioxidants, including tocopherols, ascorbic acid, rosemary extracts, lycopene and some flavonoids, should be considered as inhibitors of lipid hydroperoxide formation and subsequent degradation into aldehydic by-products [157]. These natural antioxidants are also added to foods to protect nutrients and the human body from oxidation and the generation of volatile organic compounds, especially aldehydes [158]. However, once phenolic antioxidants trap or neutralize free radical-initiated lipid peroxidation, they are also converted into reactive species, such as phenoxy radicals, which may enhance lipid peroxidation over time [159]. To prevent the accumulation of these phenoxy radicals, they are reduced back to their original compound by other phenolic compounds existing in the total plant extract as an electron donor, ensuring a continuous supply of antioxidants. The underlying mechanisms of synergistic interactions between several antioxidants are poorly understood and rarely studied [160]. Nevertheless, the most natural antioxidants used have several phenolic rings in their molecular structure and are effective in inhibiting lipid peroxidation by blocking the formation of free radicals. In addition, the increased number of hydroxyl groups, the presence of allyl carboxylic acid and the substitution of hydroxy by methoxy groups in the molecule strongly increase their reducing and antioxidant capacities [161,162]. In regard to lipid food protection strategies against oxidation, the reduction of O_2_, through airtightness and vacuum, encapsulation and light protection have been used [163,164]. Further, the strategies of antioxidant supplementation in the food industry have been successfully used to improve food shelf life. Synthetic antioxidants, such as tert-butyl-4-hydroxyanisole (BHA), 2,6-di-tert-butyl-4-methylphenol (BHT), tert-butylhydroquinone (TBHQ) and propyl gallate (PG), have been commonly added to processed foods to avoid rancidity and malodor resulting from nutrient oxidation, but their use has become limited due to growing concerns about their potential mutagenic and carcinogenic effects [165]. Therefore, the use of the total polyphenolic extract from plants allows the antioxidants to continuously quench free radicals, prevent the oxidation of biomolecules and the accumulation of oxidized phenols and the formation of secondary products, preserving the flavors, colors and texture of foods during storage [165,166]. The antioxidant capacity depends on the individual (poly)phenol content of mangosteen peel extract. The reducing capacities of their phenolic compounds can be predicted on the basis of their structural characteristics. For example, mangosteen peel extract, rich in α, β and γ-mangostin as major polyphenols with several hydroxyl and methoxy groups, is used as a potential source of natural antioxidants for prolonged application in the food industry to suppress lipid peroxidation (for structure, see Figure 2) [167]. These polyphenols can prevent some foods enriched with n-3 PUFA from quality deteriorations and lipid oxidation [168]. A study indicated that n-3 PUFA-enriched eggs were relatively stable during storage and home cooking in the presence of antioxidants [169]. N-3 PUFA ingestion without antioxidant protection results in increased plasma and urinary concentrations of hydroxyl alkanals due to peroxidized oil absorption or in vivo oxidation [170,171]. Indeed, 4-HNE and lipoxygenase (LOX)-catalyzed linoleic acid (LA) oxidation products affect the digestibility and gel properties in myofibrillar proteins from bighead carp, resulting in decreased free amino acid content in myofibrillar proteins [172]. Aldehydes can be found in many foods, including red meat, chicken, fish, dairy, grains, fruits, vegetables and alcoholic beverages. However, fried foods are particularly high in oxidized fat and can contain high concentrations of aldehydes [173].

As these lipid aldehyde compounds have adverse effects on nutrient quality and human health, it is a challenge to reduce their formation. In tissues, lipid aldehydes are metabolically removed through oxidation and reduction. They are catalyzed via the enzymes aldehyde dehydrogenase (ALDH2) and aldose reductase (AKR1B1) and are also removed by endogenous glutathione and histidyl dipeptides, such as carnosine (β-alanine-histidine). When both the enzymatic and non-enzymatic removal of lipid aldehydes is diminished, aldehyde-modified proteins accumulate in atrophic skeletal muscle during heart failure. Therefore, reducing or blocking their formation during PUFA oxidation in foods will allow for their preservation and extend the shelf life of nutrients [174].

## 10. Conclusions

Oxidative stress caused by PA exposure is the main mechanism elucidated because it can be reversed through antioxidant treatments. Other processes, such as the activation of both lipid droplets and FFARs, are gaining importance, and, therefore, several works are oriented to determine how lipid droplets are formed and which contact sites between different organelles and lipid droplets are involved in the protection against PA-induced lipotoxicity. Therefore, it is essential to use formulations that are able to effectively preserve the bioactivity of n-3 PUFA in food by adding natural antioxidants from total plant extracts to block the propagation of lipid peroxidation and to reinforce the activity of the endogenous antioxidant systems, providing protection against PUFA peroxidation and preserving foods, leading to better health conditions.

## Figures and Tables

**Figure 1 ijms-26-00788-f001:**
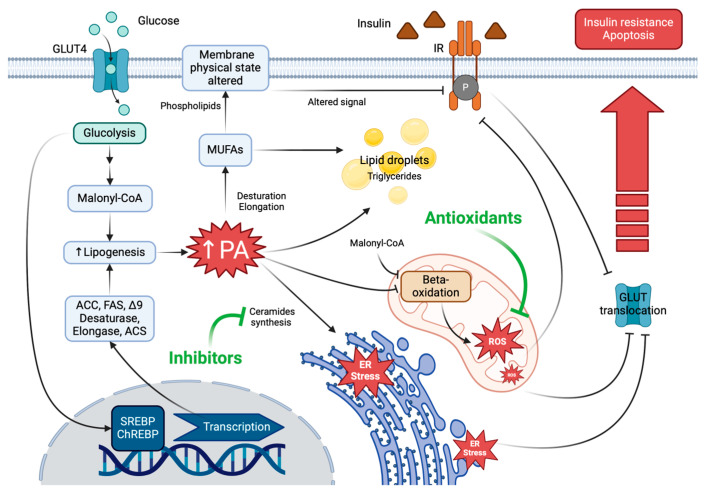
PA-induced lipotoxicity and the possible mechanism involved. Excess-carbohydrate diet increases glycolysis products and the activation of SREBP and ChREBP and, therefore, transcription of lipogenesis-associated proteins. Lipogenesis produces a high concentration of palmitic acid (PA), which will be elongated and/or unsaturated to be esterified in phospholipids, altering the composition of the membrane, or in triglycerides that will accumulate as lipid droplets. On the other hand, an excess of PA involves the inhibition of ß-oxidation as well as malonyl-CoA, leading to an increase in ROS production or in the synthesis of ceramides that produce ER stress. Alterations in the lipid composition of the membrane, the excessive production of ROS and ER stress inhibit the signaling of insulin receptors and, consequently, the translocation of GLUT receptors, promoting insulin resistance and apoptosis. These processes can be inhibited with the support of the consumption of ceramide synthesis inhibitors or natural antioxidants that prevent oxidative stress. Glucose transporter (GLUT), insulin receptor (IR), acetyl-CoA carboxylase (ACC), fatty acid synthase (FAS), acetyl-CoA synthase (ACS), sterol regulatory element-binding protein (SREBP), carbohydrate response element-binding protein (ChREBP), monounsaturated fatty acids (MUFAs), palmitic acid (PA), reactive oxygen species (ROS), endoplasmic reticulum (ER), phosphorylation (P).

**Figure 2 ijms-26-00788-f002:**
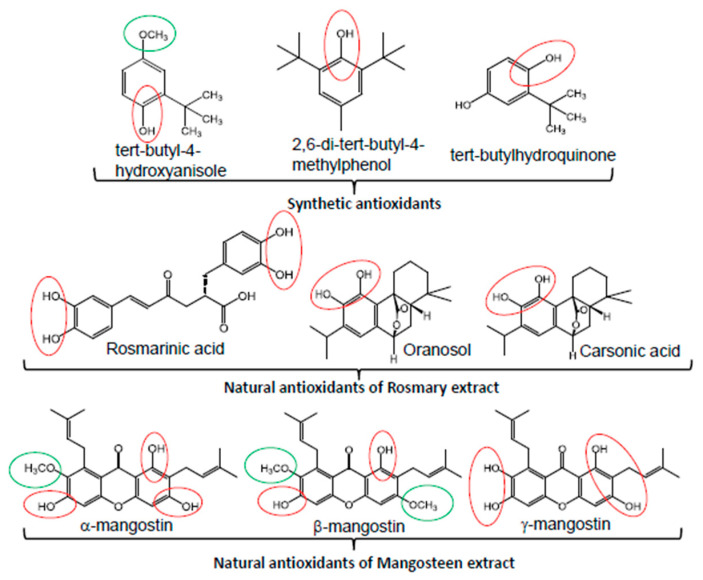
Chemical structure of synthetic and natural antioxidants used as lipid protectors. The increased number of hydroxyl groups, the substitution of hydroxy by methoxy groups and the presence of allyl carboxylic acid in the molecule strongly increase their reductions and antioxidants.

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
