# Peer review of "Protection Strategies Against Palmitic Acid-Induced Lipotoxicity in Metabolic Syndrome and Related Diseases"

_ijms, 2025, doi:10.3390/ijms26020788_

Round 1
Reviewer 1 Report
Comments and Suggestions for Authors
In this review, the authors clearly discuss a comprehensive overview of the role of palmitate-induced lipotoxicity in metabolic syndrome and related diseases. The discussion of potential intervention strategies is particularly insightful. The mechanisms of palmitate-induced lipotoxicity are well-explained, including mitochondrial dysfunction, endoplasmic reticulum stress, oxidative stress, and inflammation. The review discusses several potential intervention strategies, including endogenous protective mechanisms (Lipid Droplets, LDs), and exogenous protective effects (FFARs agonists). In addition, this review emphasizes that the protective effects of natural antioxidants like polyphenols and flavonoids against palmitate-induced damage is relevant and discuss the potential of enhancing LD formation and turnover to mitigate lipotoxicity is interesting and deserves further investigation. The manuscript is generally well-written, but there are some areas that need improvement before it can be considered for publication. Below are my detailed comments and suggestions for improvement.
1. This review mentions that palmitic acid can be converted into triglycerides by desaturation and lengthening and stored in lipid droplets, which enzymes regulate this conversion process?
2. The document mainly discusses the relationship between palmitic acid induced lipid toxicity and metabolic syndrome and obesity. Does this relationship exist in other diseases, such as type 2 diabetes and cardiovascular disease?
3. Lipid droplets play an important role in providing energy for metabolic processes and membrane biosynthesis, how do lipid droplets affect the efficiency of energy use by cells?
4. Natural antioxidants can reduce palmitic acid induced lipid toxicity, which antioxidants are most effective? What is the mechanism of action?
5. Exogenous agonist-activated FFARs have been suggested as potential therapeutic targets for the prevention of palmitic-induced lipid toxicity in beta cells. How safe and effective are they?
6. Are palmitic acid-induced lipid toxicity protective strategies clinically and in use? In the discussion section, the clinical application status of palmitic acid-induced lipid toxicity protection strategies should be supplemented.
Author Response
- Comment This review mentions that palmitic acid can be converted into triglycerides by desaturation and lengthening and stored in lipid droplets, which enzymes regulate this conversion process?
In the introduction section we mentioned that palmitic acid is stored in LD as triglyceride in adipose and non-adipose tissue without detailing the process (see lines 87 to 90). However in the section 2 the process including the enzymes involved in the conversion is added (see lines 152-158).
- The document mainly discusses the relationship between palmitic acid induced lipid toxicity and metabolic syndrome and obesity. Does this relationship exist in other diseases, such as type 2 diabetes and cardiovascular disease?
Yes of course the relation between PA-induced lipotoxicity and cardiovascular diseases and Type 2 Diabetes was commeted en the section 5 dedicated to PA and Insulin resistance and CV diseases. the manuscript already describes the relationship of palmitic acid-induced insulin resistance related to the development of diabetes and cardiovascular diseases in the section 5. Even so, we added in the same section a comment about the the relation to type 2 diabetes and cardiovascular disease in lines 429 o 436. With the focus on the lipotoxicity damage in both diseases. Thank you for the comment
- Lipid droplets play an important role in providing energy for metabolic processes and membrane biosynthesis, how do lipid droplets affect the efficiency of energy use by cells?
A comment was added to the text about this important question. See section 7 lines 526 to 534.
- Natural antioxidants can reduce palmitic acid induced lipid toxicity, which antioxidants are most effective? What is the mechanism of action?
in the sections 6 we describe the possible nechanism by which several antioxidants protect against PA induced lipotoxicity we add and edit a small conclusion paragraph highligthing the polyphenols as the most effective antioxidants related to the activation of sirtuins, Akt and AMPK as a possible mechanism (lines 508 to 510) and further investigations are needed to demonstrate the effectivety of natural antioxidants improving cellular health in models of metabolic MetS and obesity. Moreover in the section 9 we describe in general that the increased number of hydroxyl groups, the presence of allyl carboxylic acid, and the substitution of hydroxy by methoxy groups in the polyphenol molecules strongly increase their direct reducing and antioxidant capacities [161,162] (lines 611 to 613).
- Exogenous agonist-activated FFARs have been suggested as potential therapeutic targets for the prevention of palmitic-induced lipid toxicity in beta cells. How safe and effective are they?
A more detail about the safety and the affectivity of exogenous activators of FFARs was added in the text. Moreover it is well known that any synthetic product with biological activity cannot be exempt from secondary effects and more studies are need to characterize the safety . A comment was added to the section 4; lines 385 to 400
- Are palmitic acid-induced lipid toxicity protective strategies clinically and in use? In the discussion section, the clinical application status of palmitic acid-induced lipid toxicity protection strategies should be supplemented.
Although the lipotoxic effects of high-concentration of palmitic acid are well documented, the possibility that low doses of PA confer protective benefits suggests the need to explore dose-response relationships further in clinical settings. This duality could inspire dietetic recommendations and personalized therapeutic strategies. Short-term exposure to low-dose palmitic acid has been shown to improve mitochondrial function and oxidative status in cardiomyoblasts, suggesting that controlled exposure may enhance cellular resistance to mitochondrial dysfunction, to oxidative stress, inflammation and insulin resistance (doi: 10.1016/j.toxrep.2024.01.014). This comment was added to the section 5 lines 456 to 462.
Reviewer 2 Report
Comments and Suggestions for Authors
Authors review protection strategies against palmitic acid-induced lipotoxicity in metabolic syndrome and related diseases. They discuss the sources of palmitic acid, its metabolism and potential role of oxidative stress and antioxidant pathways to mitigate the palmitic acid-induced lipotoxicity.
This is an interesting review, however, several presented concepts are outdated or incorrect. Significant revision is required.
1) Authors must describe that palmitic acid is typically received with food is in esterized form. De novo synthesis also quickly esterized. Therefore, palmitic acid is not always present in potentially harmful de-esterified free form. The esterified palmitic acid can be activated by inflammation or oxidative stress which are associated with metabolic syndrome.
2) Please add that palmitic acid is a critical source of energy and a essential building block of cells. Therefore, palmitic acid is not "bad" by itself, however, metabolic conditions can lead to detrimental accumulation of palmitic acid (PMID: 38928204).
3) Correct Abstract:
The statement "PA-induced lipotoxicity through its desaturation..." is NOT correct since palmitic acid (PA) is unsaturated and cannot be desaturated.
The statement "Therefore the challenge is to avoid the oxidation of dietary unsaturated fatty acids using natural antioxidants" is unclear. Please make clear throughout the text that you discuss two completely different pathways: Lipid peroxidation (oxidative stress) and fatty acid beta-oxidation (physiological oxidation). In the statement above please refer to "lipid peroxidation".
4) Graphical abstract is not clear.
Please change the "b-Ox/ATP" box to depict a mitochondria and refer to "mitochondrial fatty acid beta-oxidation" (Panov, A.V.; Mayorov, V.I.; Dikalov, S.I. Role of Fatty Acids β-Oxidation in the Metabolic Interactions Between Organs. Int. J. Mol. Sci. 2024, 25, 12740). Please use throughout the text "mitochondrial fatty acid beta-oxidation". Do not use ß-oxidation or (ß-Ox).
5) The statement in section 3.2 is very confusing" "Mitofusin2 promotes β-oxidation of fatty acids through its direct interaction with carnitine palmitoyl transferase (Cpt1α), a fatty acid transporter into mitochondria for β-oxidation...". Please consider revision to something like this: "Mitofusin2 directly interacts with carnitine palmitoyl transferase (Cpt1α), a fatty acid transporter into mitochondria, which promotes β-oxidation of mitochondrial fatty acids oxidation.
6) "Antioxidants strategies to prevent lipid oxidation" section must be revised/corrected. The main problem is that author misrepresent the role of OH-radicals in lipid peroxidation. A) OH-radicals cannot induce lipid peroxidation because they do not reach the double bonds of fatty acid and react with any organic (saturated) C-H bond instantly (diffusion controlled reaction). B) The actual oxidants inducing lipid peroxidation includes i) hydroperoxyl radical or perhydroxyl (HO2.) and ii) Ferryl species (>Fe(IV)=O) as has been previously described (PMID: 35409406). Please remove ALL references to hydroxyl (OH) radical to avoid confusion and copying of outdated errors.
7) Please correct the formular of superoxide which is a radical anion (O2.-).
8) Add reference that all polyphenols are actually inducing expression/activation of cellular antioxidant enzymes rather than themself reacting with the free radicals through cellular systems like NRF2 where phenolic groups react with the KEAP1 to release/activate NRF2.
Author Response
- Authors must describe that palmitic acid is typically received with food is in esterized form. De novo synthesis also quickly esterized. Therefore, palmitic acid is not always present in potentially harmful de-esterified free form. The esterified palmitic acid can be activated by inflammation or oxidative stress which are associated with metabolic syndrome.
We appreciate the comment suggesting the inclusion of PA coming from food in addition to the novo biosynthesis of PA. In the section corresponding to the biosynthesis of PA, a few lines (166 to 178) were added describing lipid metabolism and the differences of palmitic acid esterification depending on its source (food or de novo synthesis) and cell type.
In this review we describe that the esterification of PA in trilglycerides as a protective mechanism against lipotoxicity however PA also acumulate cardiolipin, in ceramides and as diacylglycerol al these molecules are includes in the section 3.1, 3.2, and 3.3.
- Please add that palmitic acid is a critical source of energy and a essential building block of cells. Therefore, palmitic acid is not "bad" by itself, however, metabolic conditions can lead to detrimental accumulation of palmitic acid (PMID: 38928204).
Thank you for the suggestion, we agree with you, it seems that palmitic acid is the “bad one”. In deed we give son description about PA as a major substrate for myocardial metabolism, influencing energy production However we added a small description emphasizing its physiological activity and importance (see lines 77 to 83). Also, we incorporate the suggested citation (Ref 35) and highlight other related lines that were already in the text but could be unnoticed (lines 171 and 175).
- Correct Abstract:
The statement "PA-induced lipotoxicity through its desaturation..." is NOT correct since palmitic acid (PA) is unsaturated and cannot be desaturated.
The statement "Therefore the challenge is to avoid the oxidation of dietary unsaturated fatty acids using natural antioxidants" is unclear. Please make clear throughout the text that you discuss two completely different pathways: Lipid peroxidation (oxidative stress) and fatty acid beta-oxidation (physiological oxidation). In the statement above please refer to "lipid peroxidation".
Thank you for the observation, the word “oxidation” has been replaced for “lipid peroxidation” given a better understanding. In the case of the word “desaturation”, we think was a miss understanding, because palmitic acid is a saturated fatty acid, and the C16 unsaturated form is called palmitoleic acid. Nevertheless, the word was changed for “unsaturation” (line 32).
- Graphical abstract is not clear.
Please change the "b-Ox/ATP" box to depict a mitochondria and refer to "mitochondrial fatty acid beta-oxidation" (Panov, A.V.; Mayorov, V.I.; Dikalov, S.I. Role of Fatty Acids β-Oxidation in the Metabolic Interactions Between Organs. Int. J. Mol. Sci. 2024, 25, 12740). Please use throughout the text "mitochondrial fatty acid beta-oxidation". Do not use ß-oxidation or (ß-Ox).
We consider your observation, and the “ß-Ox/ATP” box was changed for "mitochondrial fatty acid beta-oxidation and ATP production". Thank you.
- The statement in section 3.2 is very confusing" "Mitofusin2 promotes β-oxidation of fatty acids through its direct interaction with carnitine palmitoyl transferase (Cpt1α), a fatty acid transporter into mitochondria for β-oxidation...". Please consider revision to something like this: "Mitofusin2 directly interacts with carnitine palmitoyl transferase (Cpt1α), a fatty acid transporter into mitochondria, which promotes β-oxidation of mitochondrial fatty acids oxidation.
Thank you very much for the suggestion. The sentence was changed. See line 255 to 258
- "Antioxidants strategies to prevent lipid oxidation" section must be revised/corrected. The main problem is that author misrepresent the role of OH-radicals in lipid peroxidation. A) OH-radicals cannot induce lipid peroxidation because they do not reach the double bonds of fatty acid and react with any organic (saturated) C-H bond instantly (diffusion controlled reaction). B) The actual oxidants inducing lipid peroxidation includes i) hydroperoxyl radical or perhydroxyl (HO2.) and ii) Ferryl species (>Fe(IV)=O) as has been previously described (PMID: 35409406). Please remove ALLreferences to hydroxyl (OH) radical to avoid confusion and copying of outdated errors.
Thank you very much for this comment. Indeed we describe the classical pathway that involves OH-radical which result from fenton reaction of H2O2 . However we have decided to adopt your proposal that the reactive oxygen species that can promote lipid peroxidation is the perhydroxyl radical, which arises from the protonotion of superoxide anion. The fact that the hydroxyl radical has a very shortl life (10-9 s) and that its formation requires the presence of iron or copper. We include the two references DOI: 10.1134/S0026893318020097 and doi: 10.3390/ijms23074047. (See lines 592 to 594)
- Please correct the formular of superoxide which is a radical anion (O2.-).
The formulae of superoxide is corrected in all the text.
- Add reference that all polyphenols are actually inducing expression/activation of cellular antioxidant enzymes rather than themself reacting with the free radicals through cellular systems like NRF2 where phenolic groups react with the KEAP1 to release/activate NRF2.
A comment about phenolic compound activating the system Keap-NRF2 to protect against toxicity is added see line see line (522 to 528). Sreference ias also added (see ref 133).
Round 2
Reviewer 2 Report
Comments and Suggestions for Authors
This is a revised review on the protection strategies against palmitic acid-induced lipotoxicity in metabolic syndrome and related diseases.
The authors have substantially revised the text and Figures in response to reviewers' comments and critic.